



# Design optimisation of an open-source reference rotor library for multi-rotor development and innovation

Abdirahman Sheik Hassan[1], Neha Chandarana[1], Rainer Groh[1], and Terence Macquart[1]

[1]Bristol Composites Institute, School of Civil, Aerospace and Design Engineering, University of Bristol, Queen's Building, University Walk, Bristol BS8 1TR, United Kingdom

**Correspondence:** Abdirahman Sheik Hassan (a.sh.2019@bristol.ac.uk)

**Abstract.** The multi-rotor wind turbine (MRWT) is an emerging concept that promises benefits in aerodynamic, structural, economic and environmental performance over the existing single-rotor paradigm. However, research into MRWTs generally lacks detailed aero-servo-elastic design studies, and modelling efforts undertaken in recent years have mainly focussed on rotor interaction, wake mechanics and support structure optimisation, without detailed modelling of the rotor blades themselves.

Partly due to the lack of small (<750 kW) reference rotor models, in the limited cases where rotors are modelled, blade designs are typically derived by geometrically scaling existing reference rotors. However, the choice of reference models and the underlying assumptions used for scaling can result in rough and non-optimised aerodynamic and structural blade designs, and significant variations in rotor performance — making fair and transparent comparisons between multi-rotor configurations, and against single-rotor configurations, very challenging. A roadblock in the comprehensive design of MRWTs therefore lies in

the lack of bespoke blade and rotor models suitable for the MRWT use case. To address this limitation, we have developed an open-source library of reference blade and rotor designs, covering a rated power range of 100 kW–1 MW, which have been generated using an in-house aeroelastic optimisation software, ATOM. While the generation of these novel rotor designs has been motivated by their application to multi-rotor research, their use is not strictly limited to this context. The newly designed rotors are compared against existing reference models, showing good agreement in mass and aerodynamic properties.

Moreover, we propose and demonstrate an interpolation procedure to generate intermediate rotors within the rotor library for custom multi-rotor models, enabling the generation of MRWT models of any overall power rating, using any number of rotors. This allows for comparative studies between MRWTs and greater exploration into multi-rotor scaling laws. In this paper, the design and optimisation process is discussed in detail, with the resulting designs shared as open-source models in the OpenFAST format, alongside the comparison of the aeroelastic responses predicted by ATOM and OpenFAST. The repository

of rotor designs can be found at https://github.com/Abdi-SH/MRRL_OpenFAST_Files.

## 1   Introduction

As the mounting climate crisis continues to highlight the need to transition to clean energy generation, the demand for installed wind power capacity is growing. The 2025 global wind report from the Global Wind Energy Council predicts an additional 982 GW to be installed in the next five years (Council, 2025). In an environment of accelerated growth, the role of the research





community in developing new turbine technology to enable lower costs and more robust installations cannot be understated. A common metric used to assess the impact of new technologies in the context of wind turbines is the levelised cost of energy (LCoE) — a measure of the costs of a power plant per unit energy produced over its lifetime. One of the primary trends driving down this figure in previous years has been the improvement of the power capacity of individual turbines through the increase in the size of rotors, resulting in rotor diameters exceeding 300 m (Council, 2025). However, this method of upscaling has resulted in a wide array of technical challenges. These include difficulties in manufacturing large composite structures to a high degree of quality and consistency (Veers et al., 2023), increased costs of transportation and installation (Peeters et al., 2017), and difficulty in addressing the high environmental impact of producing and disposing of high-grade composite materials (Beauson et al., 2022). More fundamentally, the inherent scaling properties of single-rotor (SR) wind turbines call into question the sustainability of their continued upsizing. While the power output of a turbine increases with the blade length squared, blade mass (and therefore cost) increases with length cubed according to the commonly cited "square-cube law" (Jamieson and Branney, 2012), or to an exponent between 2.1 and 2.9 according to historical trends (Crawford, 2007). The risk of blade costs growing faster than the associated power increase therefore undermines the ability of the wind industry to continue decreasing the LCoE of wind turbines by this method.

The concept of the multi-rotor wind turbine (MRWT) is a potential solution to a number of the issues wind turbine designers are facing in the drive to further reduce LCoE and increase the power density of individual turbine installations. By replacing a single rotor with a set of smaller rotors in an array on a single support structure with equal overall rated power, rotor mass may be reduced by a factor of up to $\frac{1}{\sqrt{n}}$, where $n$ is the number of rotors in the MRWT (Jamieson and Branney, 2012). An example of this concept is illustrated in Figure 1. Smaller individual rotors are easier to transport, manufacture at a higher volume, install, and repair, offering the ability to use existing rotors "off-the-shelf" and provide redundancy such that the turbine can continue operating in the case of individual rotor failures. Interaction between individual rotors has been suggested to increase total power output, with experimental and simulation studies showing potential gains of up to 8% depending on the rotor number (Chasapogiannis et al., 2014). The rotor interaction has the additional effect of reducing the wake recovery distance when compared against single rotors of the same rated power (Van Der Laan and Abkar, 2019). Despite the purported benefits of the multi-rotor concept, empirical data are sparse, and detailed design studies based on comprehensive aeroelastic optimisation are non-existent (Sheik Hassan et al., 2026).

As a consequence of the conventional SR design paradigm, research in wind turbine blade modelling and design has been geared towards large blades for multi-megawatt single-rotor applications (Veers et al., 2023). The smallest (detailed and open-source) reference wind turbine known to the authors has a rating of 750 kW, which has led some MRWT design studies to scale down existing, and occasionally outdated, reference models using approximate scaling exponents to compare against SRs of equal total rated power (Jamieson, Branney, and Hart, 2017; Ferede and Gandhi, 2022). Relying on simple geometric scaling may be problematic when differences in blade performance for different Reynolds number operating ranges are not taken into consideration. This is especially the case with downscaling across large differences in rated power, negatively affecting the lift and drag performance of the downscaled blade. Geometric scaling may also result in inaccurate mass predictions for scaled blades as the resulting blade may be over- or under-designed for the structural loading experienced at the new scale. In the case





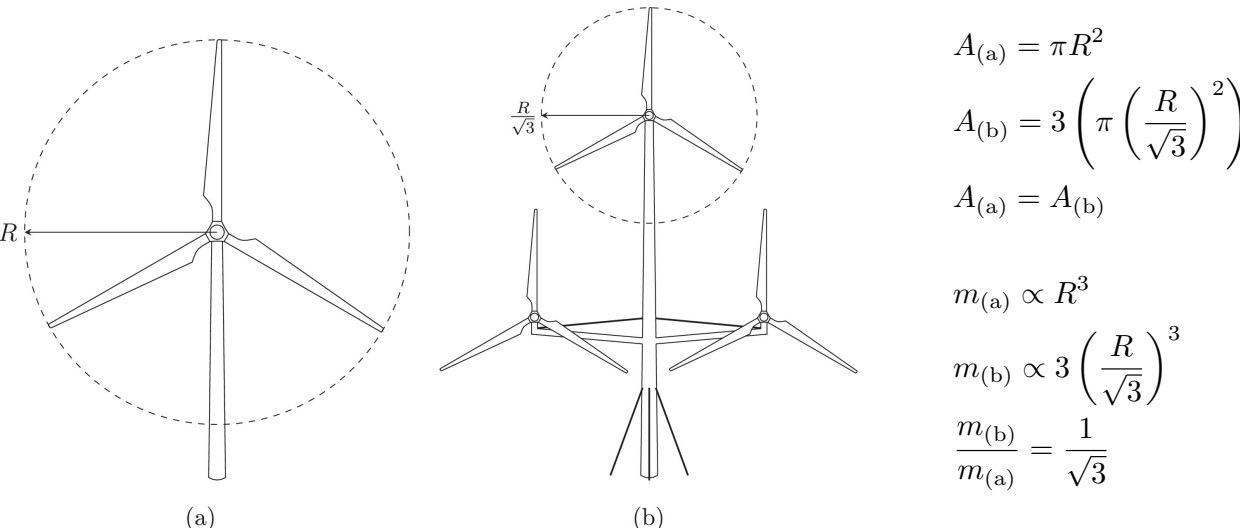

**Figure 1.** Sizing of a tri-rotor MRWT by dividing the swept area $A$ of (a) a single rotor into (b) three smaller rotors, thereby reducing the total rotor mass $m$ without sacrificing power. To the right of the figure, the equivalence in area and the mass saving based on cubic blade scaling (exponent $n = 3$) are shown. Reproduced with permission (Filsoof et al., 2021).

of comparing SR and MRWTs using existing reference models, differing power curves can lead to unrepresentative results. At below-rated speeds, different reference turbines produce different fractions of their rated power, resulting in discrepancies in performance. Restricting studies to existing reference models also restricts the multi-rotor arrangements that can be explored based on the models available to the research community. On a conceptual level, downscaled reference rotors are not designed with the benefits of reduced blade size in mind, thereby foregoing the opportunity to use sustainable materials, tailor the design

to MR dynamics, or to simplify the design for higher manufacturing volume. Without these considerations, a comprehensive comparison between SR and MRWTs cannot be achieved as the benefits of the MRWT concept are not fully exploited.

This work is therefore concerned with the design of a range of small rotors through aero-servo-elastic optimisation designed for, but not limited to, the context of multi-rotor systems. The designs range from a rated power of $100\,\text{kW}$ to $1\,\text{MW}$ as a starting point to address the gap in available models and to compare against existing smaller rotor models. The goal of this

work is to share the designs as open-source reference models for the research community. The format for the open-source blade models is chosen to be compatible with the OpenFAST software from NREL (NREL, 2025). As the main intended use of these blade models is for the design and analysis of multi-rotor systems for comparison against SRs and other MRWT designs, an additional aim is to develop a procedure to interpolate between the designs to arrive at an intermediate rated power. This facilitates the generation of multi-rotor models with the same overall rated power, but different rotor numbers,

allowing for further characterisation of blockage effects as well as serving as a stepping stone for the optimisation of rotor number and arrangement. While the interpolation is in itself a form of geometric scaling, the process takes place within small





ranges of power ratings and is validated against an optimised blade at the same rating in Section 3.3. This paper details the design process for the library of reference rotors, demonstrates the aerodynamic and structural performance of a selected rotor model, and verifies the functionality of the rotor interpolation procedure. Details regarding open access of the rotor models

are shared in the conclusion. This work is primarily concerned with the aero-structural design of the wind turbine blades, while the surrounding rotor parameters (masses and dimensions of the nacelle, hub, generator, etc) are defined using scaling assumptions based on empirical data. This data is also made available in the associated repository. It should be noted that the designs detailed in this study are not intended to be "ready-to-build" high-fidelity models, but rather starting points to be used in multi-rotor research for the purpose of iterating towards optimum system designs. The detailed design process is provided

for these reference models in the following sections for the purpose of allowing for rigorous reproduction of results. As such, further refinement is encouraged through the open-source dissemination of the rotor library.

## 2  Methodology

### 2.1  Initial Design

For consistency, the design process for each reference blade is identical across the range of power ratings. The rated windspeed

is used in conjunction with the rated power to arrive at a rotor radius, using empirical formulae to generate an initial twist and chord distribution. The main focus when creating the initial designs is the generation of a structurally conservative blade design that is a feasible starting point for the optimisation process. Hence the initial geometry and control parameters are rough starting estimations made in order to begin the design process. Equation (1) displays the turbine power formula used to select blade length,

$$P = \frac{1}{2}\rho V^3 A C_{\mathrm{P}}, \tag{1}$$

where $P$ refers to extracted aerodynamic power, $\rho$ to air density, $V$ to windspeed, $A$ to rotor area and $C_{\mathrm{P}}$ to the power coefficient. At this stage, rated speed $V$ is set to 11.5 m/s, design $C_{\mathrm{P}}$ as 0.45, and air density $\rho$ is assumed to be the standard sea level value of $1.225\,\mathrm{kg/m^3}$. Values in this range are commonly used for preliminary rotor sizing, although the $C_{\mathrm{P}}$ used is on the lower end (Hasan, El-Shahat, and Rahman, 2017; J. Jonkman et al., 2009). The rated windspeed and design power coefficient

lead to a fixed rotor power density across the library range of approximately 420 W/m$^2$, which is consistent with values given for smaller rotors in the study range (Larsen and Voelund, 1998).

The initial spanwise chord distribution is determined via optimum rotor theory as presented in (Liu, Wang, and Tang, 2013). This approach uses the design lift coefficient $C_{\mathrm{L}}$ and tip speed ratio $\lambda_0$, as well as the rotor radius $R$ and number of blades $B$. These values are then used to calculate chord length at the $i$th blade section with a spanwise location location of $r_i$. The local

tip speed ratio is calculated as:

$$\lambda_i = \lambda_0 \left( \frac{r_i}{R} \right), \tag{2}$$

which is then used to calculate the local relative inflow angle





$$\phi_i = \left(\frac{2}{3}\right)\tan^{-1}\left(\frac{1}{\lambda_i}\right). \tag{3}$$

The $\frac{2}{3}$ factor encodes the optimal Betz condition of axial induction factor $a = \frac{1}{3}$, and zero tangential induction factor. The inflow angle is used to calculate the optimal local chord width

$$c_i = \frac{8\pi r_i}{BC_{Li}}(1 - \cos(\phi_i)). \tag{4}$$

For all blades designed using this method, their respective rated windspeeds and rotational speeds (set by the max tip speed limit) are used to calculate the design tip speed ratio. The design lift coefficient is set to 1.1 for the inboard sections and 1.3 for the outboard sections. Chord distributions are generated for lift coefficient values ranging from 0.5 to 1.5, with the final values chosen to give a balance between minimising blade mass and ensuring aeroelastic stability. The resulting chord distribution is structurally feasible (i.e. passes structural checks) without the increased mass associated with using lower design $C_L$ values. The initial twist distribution is determined through the following formula provided by (Habali and Saleh, 2000):

$$\beta_i = \left(\frac{R\alpha_t}{r_i} - \alpha_t\right) - k\left(1 - \frac{r_i}{R}\right), \tag{5}$$

where the initial twist angle $\beta_i$ at station $i$ is determined as a function of the ratio between its spanwise location $r_i$ and the total length $R$, as well as the $\alpha_t$ (tip angle of attack) and a tuning constant $k$ (left at 1 for this case) . The tip angle of attack is calculated as the sum of the inflow angle $\phi$ and the tip twist $\beta_t$, minus the zero lift angle of attack $\alpha_0$. The inflow angle in this case uses the design windspeed (7 m/s) rather than the rated speed used in the chord calculation, giving an initial twist distribution closer to that of the nearest existing reference wind turbines (Rinker and K.Dykes, 2018). $\beta_t$ and $\alpha_0$ are set to -2 and -3 degrees respectively. The airfoils used for the WindPACT and NREL 5 MW reference blades are both considered for utilisation in this project, and are evaluated against a scoring system judging the structural and aerodynamic properties of each airfoil (Rinker and K.Dykes, 2018; J. Jonkman et al., 2009). These include thickness to chord ratio, lift to drag ratio, proximity of the airfoil Reynolds number to the value experienced at the proposed blade station, and the smoothness of the post-stall behaviour. The NREL 5 MW airfoils are chosen for their superior lift-to-drag ratios, after confirming with XFOIL (Drela, 1989) that their aerodynamic performance in the attached flow region is not majorly hindered by their application to a smaller Reynolds number range. The scoring system and the XFOIL results are provided in the associated repository.

The initial skin thickness distributions are taken from the DTU 10 MW reference wind turbine blade to provide a conservative starting point for the structural optimisation process (Bak et al., 2013). The blades consist of glass fibre sandwich panel with a balsa core. The material layup from the DTU reference design is maintained throughout the optimisation process, with only the relative thicknesses of the layers being active design variables. Rotors with a rated power of 500 kW and above had initial skin thicknesses based on half of the DTU 10 MW distributions, while ratings below 500 kW begun with a quarter of the same reference values. The construction of the blade follows a box-beam approach, with the additional constraint of a straight web profile through the thickest portion of the blade to facilitate higher volume production. This "web-straightening" adjustment is applied to the initial design and in the post-processing phase and is intended to improve manufacturability. Figure 2 displays





visually the straight web criterion enforced in the design process. The choices in the initial design variables prior to optimisation
have some bearing over the achievable optimum design, as is discussed in Section 3.1.

## 2.2 Sizing Optimisation Method

ATOM is used to optimise the distributions of chord, twist, thickness-to-chord ratio and material layup thicknesses to minimise
the cost of energy (Macquart et al., 2020). The software uses the globally converging method of moving averages technique
(GCMMA), which is a sequential gradient-based optimisation method (Svanberg, 2007). For each objective evaluation, the
blade model is simulated under design load cases (DLCs) 1.1 and 1.3 defined in IEC 61400 standards to capture normal op-
eration and extreme turbulence respectively (International Electrotechnical Commission (IEC), 2019). As the designs detailed
in this study are not intended as "ready-to-build", the full suite of twenty-two DLCs defined by the IEC 61400 standards are
condensed to just DLCs 1.1 and 1.3 due to their roles as critical design driving cases (Bay et al., 2019; Robertson and J. M.
Jonkman, 2011). Simulated windspeeds range from cut-in to cut-out in order to determine the rotor performance in the full
range of operation. The design variables for the study are summarised in Table 1. Aerodynamic design variables are defined
by 8 spanwise control points, which can either be active (editable by the optimiser), inactive, or set to the value of the previous
control point. These variables are then edited by adjusting the control points of a spline curve added on top of the original
distribution, as illustrated in Figure 3. This approach allows for the baseline design to be defined exactly as intended regardless
of the number of design variables used to model it. As shown in Figure 3, a large range of distributions can be achieved with a
relatively small number of design variables using this method. For the chord distribution, the bounds are larger in the negative
direction to give the optimiser room to reduce chord as a means to minimise mass, given that the initial design is structurally
conservative. Structural design variables are separated by material and region, with 11 spanwise control points each for the
skin thicknesses and 14 for the webs. An illustration of the regional divisions as well as an example thickness distribution are
given in Figure 4. The root chord, twist and thickness-to-chord ratio values are fixed in order to account for lack of accuracy
in the root model, as well as enforcing a straight cylindrical section in the root for the hub joint. The tip values are set to either
the same as the previous value or fixed, to prevent large changes at the tip. In each of these cases, the design variable count
in Table 1 does not include fixed design variables. For the material thickness variables, the lower bound is a single composite
layer at $8 \times 10^{-4}$ m, while the upper bound is defined as 10% of the profile thickness at that cross-section. This applies to both
the upper and lower surfaces of the blade cross-section, as the material distribution is modelled as symmetrical about the chord
axis. The ability of the optimiser to converge to an optimum design is not entirely independent of the initial values of the design
variables, especially when values at the root and tip are fixed. This results in the potential to run into local minima, which is
discussed further in Section 4.

Constraints are enabled to ensure the structural, aeroelastic and manufacturing feasibility of the blades. Buckling failure
indices are calculated according to analytical panel buckling (Lindenburg, 2005), while the failure strain is calculated as the
lowest strain to produce any of the failure mechanisms modelled, including fibre compression, tension, and inter-fibre failure.
The failure index calculations used by ATOM are described in greater detail in (Scott et al., 2020). Blade designs with a high
degree of bend-twist coupling were found to cause discrepancies in the validation stage (Section 3.2) due to the differing



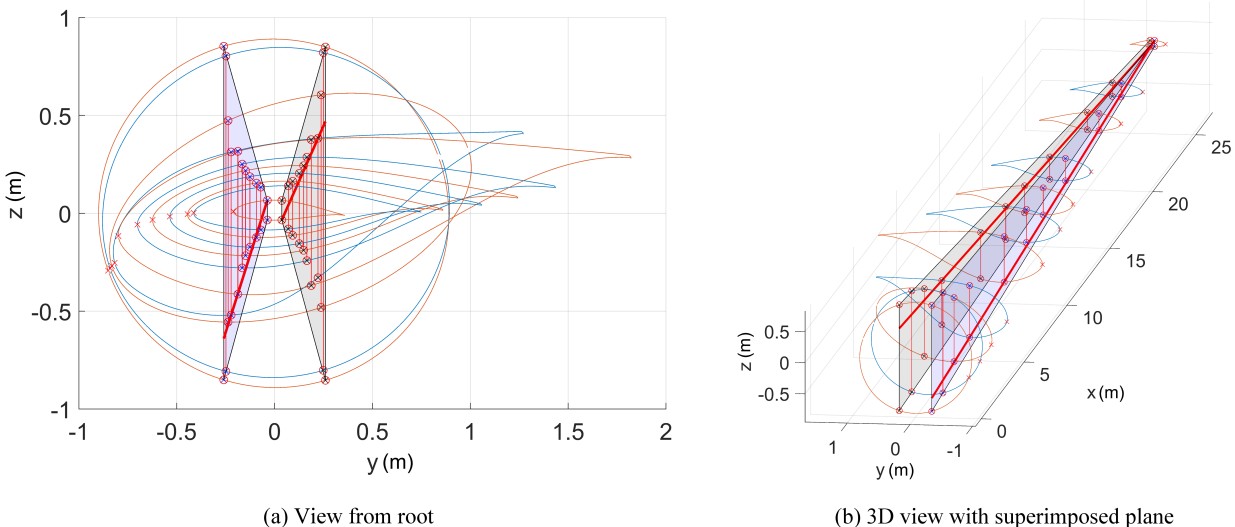

(a) View from root · (b) 3D view with superimposed plane

**Figure 2.** Web straightening algorithm output for the 1 MW initial design. The algorithm adjusts the connection points of the webs by removing twist and checking the proximity of the web against a straight projected plane. In this case, the webs are constrained onto the planes passing through the webs at the tip and webs of the section with the widest spar cap.

**Table 1.** Summary of design variables.

| Design Variable (DV) | Number | Bounds [Lower, Upper] | Comments |
|---|---|---|---|
| Chord | 5 | [0.1748 m, 1.5 m] | First two root DVs fixed, tip same as previous |
| Twist | 5 | [-30°, 15°] | First two root DVs fixed, tip same as previous (positive nose-up) |
| Thickness to chord ratio | 5 | [0.18, 1] | First two DVs fixed, tip fixed |
| Material thicknesses | 180 | [1 layer, 10% of profile] | All layer thickness DVs |

structural models used in ATOM and the validation code (openFAST). For this reason, a constraint is set on the coupling of the blades, as defined by the ratio of dominant to non-dominant degrees of freedom in each eigenmode, to not increase from the baseline design by more than 20%. In this case, the eigenmodes refer to the result of the modal decomposition performed on the linear finite element beam model of the blade (Macquart et al., 2020). Aside from ease of manufacture, the web straightening algorithm also serves the function of reducing coupling by placing the pitch axis of the blade along the thickest portion of each aerofoil. As this algorithm is not applied within the optimisation loop, the baseline "web-straightened" design is used as a benchmark for satisfactory coupling. Table 2 provides a summary of these constraints.

Sensitivity studies are also carried out in this work to add robustness to the design method. First, the rated windspeed and associated blade length are varied to study the trade-off between annual energy production (AEP) and blade mass. This results in two alternative designs for the 100 kW blade optimised for 10.5 and 12.5 m/s, respectively, both of which are provided in the



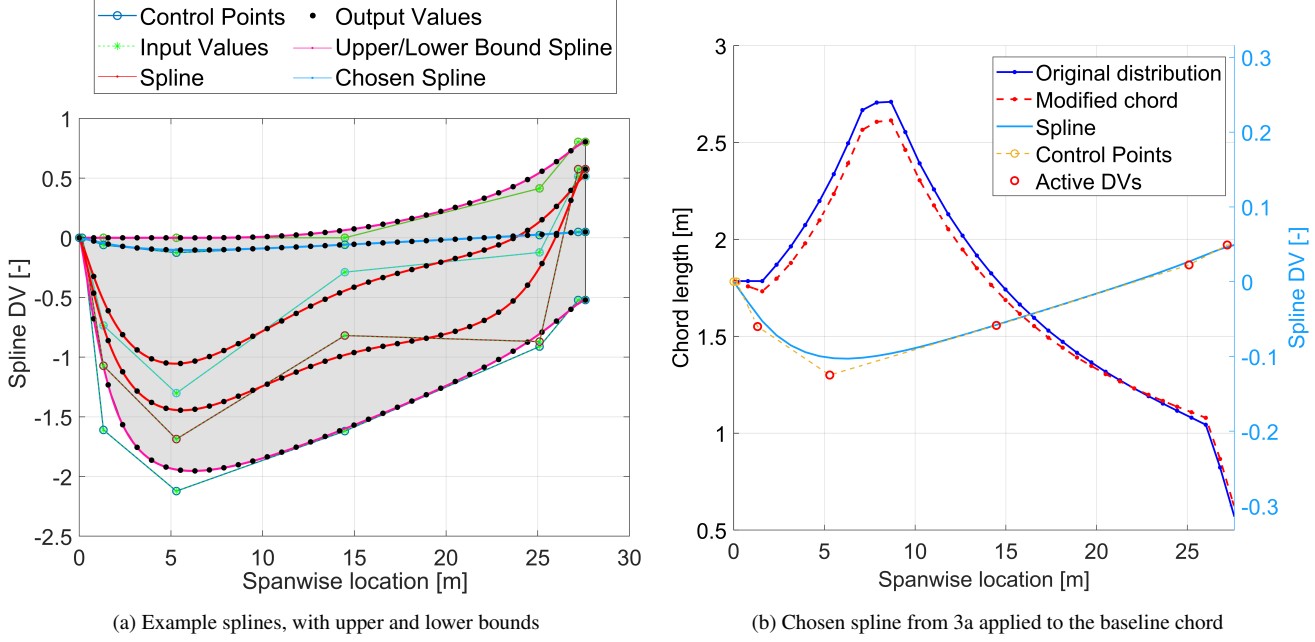

(a) Example splines, with upper and lower bounds

(b) Chosen spline from 3a applied to the baseline chord

**Figure 3.** Illustration of "baseline plus spline" design variable adjustment process as applied to the chord distribution of a 1 MW blade design. (a) displays the range of spline curves that can be generated based on the design variable bounds (shaded in grey), and (b) shows the baseline and modified chord distributions following the implementation of the blue spline curve highlighted in (a).

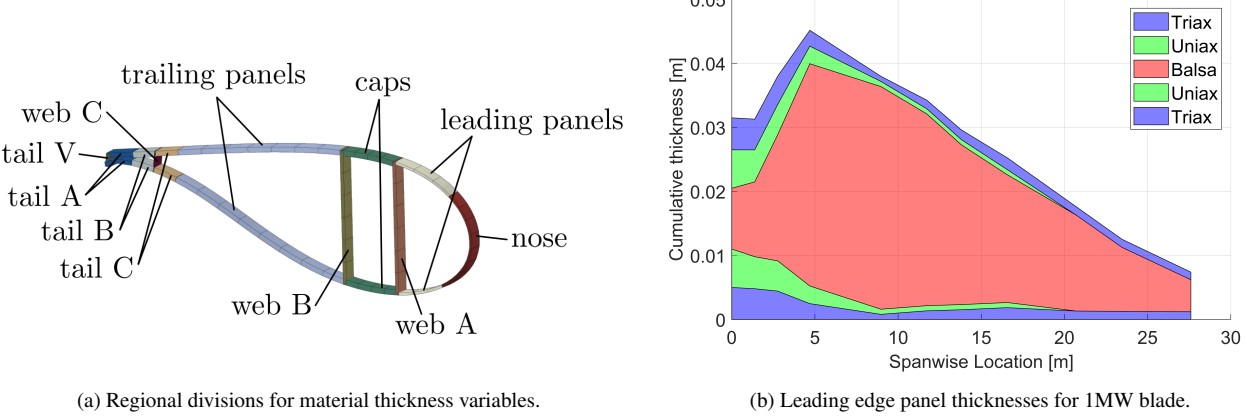

(a) Regional divisions for material thickness variables.

(b) Leading edge panel thicknesses for 1MW blade.

**Figure 4.** Depiction of material thickness DVs, along with an example of the leading panel thicknesses from the 1 MW design. The thickness of each layer in each region is set as an active design variable. (a) is reproduced with permission from (Bak et al., 2013).





**Table 2.** Summary of optimisation constraints.

| Constraint | Value | Comments |
|---|---|---|
| Tip Speed | < 80 m/s | Maximum tip speed limit based on noise limits (J. Jonkman et al., 2009) |
| Buckling | < 0 | Negative buckling failure index - safety factor 1.35, reserve factor 3.46 |
| Strength | < 0 | Negative strain failure index from Puck criterion - safety factor 1.35, reserve factor dependent on material/failure type as per (Scott et al., 2020) |
| Stability | < -0.001 | Linearised aero-servo-elastic rotor model has no eigenvalues with a positive real part, with a margin of error of 0.001 |
| Maximum deflection | < 10% | Translational deflection constrained to less than 10% of blade length |
| Fatigue | < 0 | Negative fatigue failure index based on linear Goodman analysis (Scott et al., 2020) |
| Blade coupling ratio | < 1.2 | Coupling cannot increase by more than 20% from the baseline design |

final design library for use in further study. The remaining rotors are sized for a rated speed of 11.5 m/s to be consistent with the power densities of similar existing reference blades (Rinker and K.Dykes, 2018).

## 2.3 Optimisation Objective

The second sensitivity study concerns the optimisation objective. The existing cost model implemented in the ATOM software is not suitable for the scale of blades investigated in this study; the creation of such a model is proposed for future work. The INNWIND cost model (Chaviaropoulos et al., 2014) used in ATOM was created with the goal of upscaling into multi-megawatt turbine ratings, and calibrated using data from large (5 MW +) turbines; hence, its validity for use with smaller rotors is questionable. For example, using the INNWIND cost model to estimate the nosecone/spinner mass of the rotors in the rated power range discussed in this work results in negative mass predictions. To account for this, we use a linear combination of the blade bill of materials (BoM) and annual energy production (AEP), with a sensitivity study carried out on the relative weighting of each parameter to find the objective that leads to the maximum AEP with minimum blade mass. The optimisation problem is therefore posed as follows:

$$
\min_{\mathbf{x}} \quad f(\mathbf{x})
$$

$$
\text{subject to} \quad g_i(\mathbf{x}) \le 0, \ i = 1, \dots, m.
$$
(6)

where $\mathbf{x}$ is the vector of design variables and $g_i$ is the $i^{th}$ constraint function from Table 2. The objective for each iteration $j$ is set in the following form:

$$
f_j = \alpha \left( \frac{AEP_{\text{baseline}}}{AEP_j} \right) + \beta \left( \frac{BoM_j}{BoM_{\text{baseline}}} \right).
$$
(7)





In Equation 7, the subscript "baseline" refers to the value derived from the initial design, meaning that the performance
improvement is encoded in the objective function value. The weightings $\alpha$ and $\beta$ are selected based on the relative "contribution" of the AEP in the objective function. This contribution is determined by calculating the change in the overall objective as caused by a 20% perturbation in each parameter. In this case, a relative contribution of 5 means that when AEP improves by 20%, the objective is reduced by $5\times$ more than the corresponding reduction resulting from a 20% improvement in BoM. In terms of partial derivatives, AEP having a relative contribution of 5 can be described as:

$$\mathbf{A} = AEP_j/AEP_{\text{baseline}}$$
$$\mathbf{B} = BoM_j/BoM_{\text{baseline}} \tag{8}$$
$$\frac{\partial f_j}{\partial \mathbf{A}} = 5\frac{\partial f_j}{\partial \mathbf{B}}$$

Combinations of constants $\alpha$ and $\beta$ are generated for relative AEP contributions of 0.2 to 100, as displayed in Table 3. Below a value of 1, the BoM has a greater influence on the objective value than the AEP.

**Table 3.** Weightings of AEP and BoM as implemented in Equation 7 to give a range of relative contributions of AEP.

| Relative Contribution of AEP | AEP Weighting ($\alpha$) | BoM Weighting ($\beta$) |
|---|---|---|
| 0.2 | 2.85 | 12 |
| 1 | 12 | 10 |
| 5 | 12 | 2 |
| 10 | 12 | 1 |
| 40 | 12 | 0.25 |
| 70 | 12 | 0.142 |
| 100 | 12 | 0.1 |

The most suitable balance of weightings is determined by applying each combination of weightings to the optimisation of a 100 kW blade, and analysing the performance of each design. Figure 5 shows the results in AEP and BoM performance of blades designed using each objective. An increase in the relative contribution of AEP has little effect on the final achievable AEP, as the difference between the AEP with the highest and lowest contributions is approximately 0.25%. Giving the BoM a greater influence in the objective allows the optimiser to reduce the mass of the blade without a significant loss in AEP. This is the case up until the BoM is weighted equally or more strongly than the AEP, resulting in a relatively large AEP loss being traded for a small reduction in mass. As a result of this study, the combination of weightings giving a relative AEP contribution of 5 is selected for all subsequent optimisations.

## 3   Results

In this section, the performance of the custom reference rotors as predicted by ATOM is presented. As the rotor models are made available in an open-source format through NREL's OpenFAST software, the performance predictions from each program are



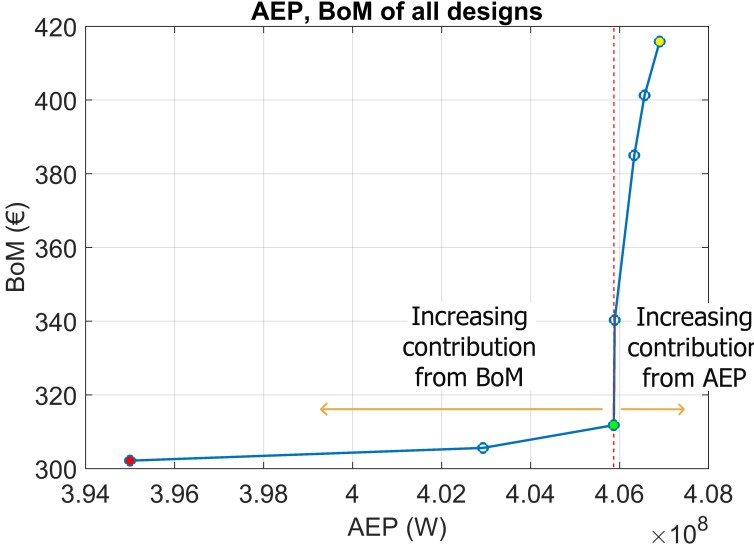

**Figure 5.** AEP and BoM performance of resulting designs from optimisation objectives with differing weightings shown in Table 3. Three points/objectives are highlighted. The red point has a relative AEP contribution of 0.2, $\alpha = 2.85$ and $\beta = 12$. The yellow point has a relative AEP contribution of 100, $\alpha = 12$ and $\beta = 0.1$. The green point has a relative AEP contribution of 5, $\alpha = 12$ and $\beta = 2$, and corresponds to the objective chosen for this study.

compared against one another for validation. To benchmark the design process carried out with the ATOM software, one of the
custom optimised blades is designed for a rated power of 750 kW to coincide with the smallest publicly available reference wind turbine, the WindPACT 750 kW turbine (Rinker and K.Dykes, 2018). The resulting blade design achieves similar aerodynamic properties although the custom blade mass was heavier (2,410.5 kg compared to WindPACT's 1,941 kg, including the mass of the root connection in both models). The WindPACT model experiences also a slightly greater spike in rotor thrust at rated speed. These discrepancies are illustrated in Figure 6.

Figure 7 shows an overview of the reference blade models resulting from this study. Over the range of power ratings generated, the fitted power law predicts a scaling between rotor radius and blade mass to the power of 2.43 (Figure 7b), which correlates to existing literature reporting scaling indices of 2.1–2.9 (Jamieson and Branney, 2012; Crawford, 2007). This is significant as the cost saving benefit attributed to the multi-rotor concept through beneficial scaling relies on the blade mass scaling with the blade length to a power greater than 2. In this case, the result is sensitive to the degree of optimisation of each
individual blade in the design library; however, matching historical trends adds confidence to the validity of the rotor designs detailed in this study.





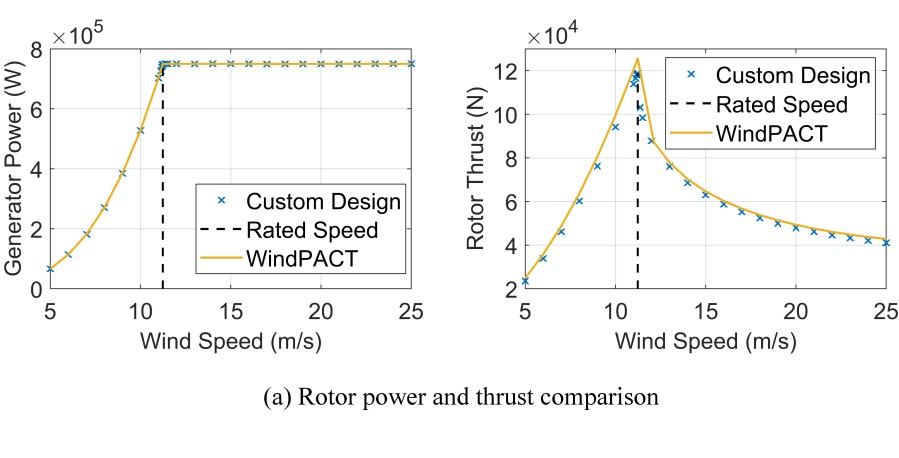

(a) Rotor power and thrust comparison

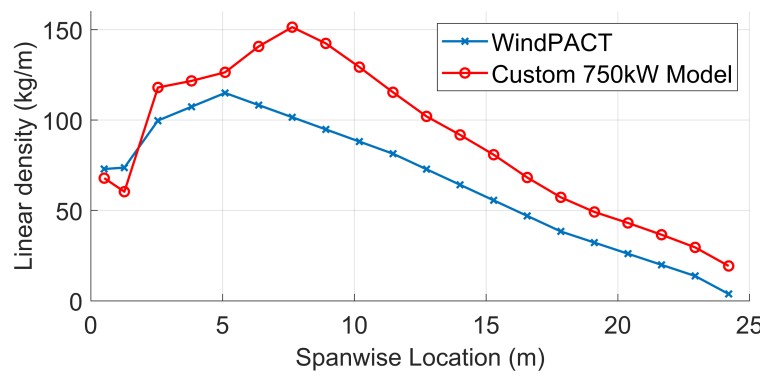

(b) Cross-sectional blade mass distribution

**Figure 6.** Comparative mass and aerodynamic properties between our optimised 750 kW rotor and the WindPACT 750 kW reference turbine.

## 3.1 Rotor Performance

The performance assessment for the rotor library focuses on the extreme ends of the design range, namely the 100 kW and 1 MW designs, after being optimised for the LCoE objective described in Equation 7. Each design is optimised for a minimum

of nine iterations, which is found to be sufficient to achieve convergence to a point beyond which attempts by the optimiser to continue reducing the objective function results in constraint violations. Testing convergence over a larger number of iterations was not considered due to the associated computational cost, meaning that the designs may not be fully converged. However, the level of convergence achieved at nine iterations is deemed sufficient as a starting point for this design library. Small violations of constraints during the optimisation are tolerated, given that each optimised design is then put through an additional post-

processing step to further reinforce areas experiencing structural failure. This approach ensures that all final designs are fully feasible according to the constraint list in Table 2. The optimiser convergence history for the final optimised 100 kW and 1 MW



WIND
ENERGY
SCIENCE
DISCUSSIONS

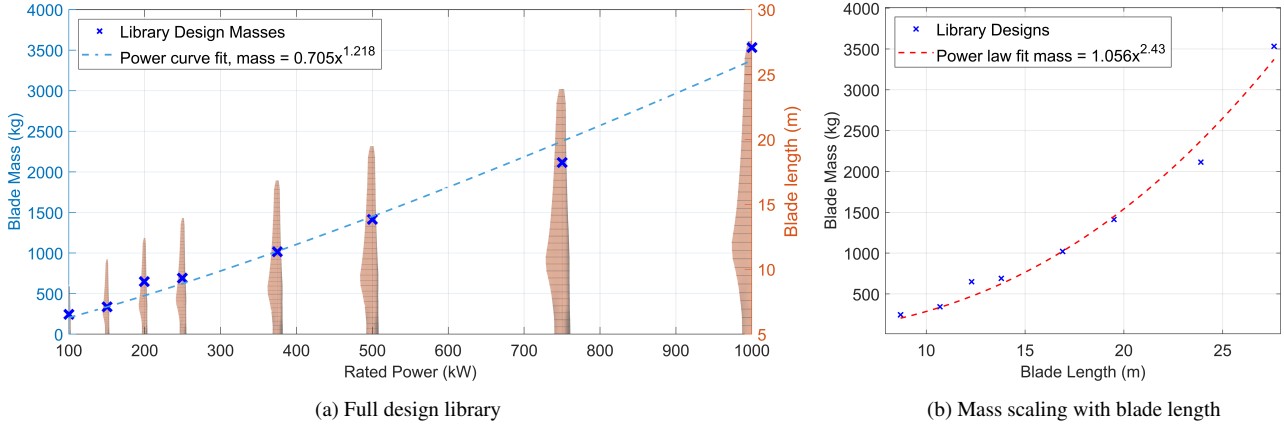

(a) Full design library    (b) Mass scaling with blade length

**Figure 7.** Blade mass and length scaling across the rotor design library. (a) shows the blade mass against power rating for each rotor design, with a power law fit. The overlaid blade models correspond to the blade lengths/rotor radii measured on the right-aligned $y$-axis. (b) shows the scaling of the blade masses in the rotor library against the blade length, with the power law predicting a scaling exponent of 2.43. The masses displayed are the direct outputs of the optimisation, without any post-processing.

designs are given in Figures 8. In the 100 kW case, the strain failure limit is the driving constraint, while for the 1 MW model it is a combination of strain and fatigue failure indices.

The aerodynamic characteristics for the 100 kW and 1 MW designs are displayed in Figure 9 and Figure 10. Included in
these figures are the power and thrust characteristics from the rotor cut-in to cut-out windspeed. The results are derived from quasi-static aeroelastic simulations at each windspeed. The trends follow the conventional appearance of a variable-speed, pitch-controlled wind turbine rotor, for which the generated power is ramped up to the rated windspeed, before pitching out of the wind and shedding power to limit thrust loading. In this case, ATOM computes the optimal control scheme prior to running the set of quasi-static aeroelastic simulations, and uses the pre-computed scheme to select a rotor speed and pitch angle for the
relevant simulation. The distribution of sampling points differs slightly between the 100 kW and 1 MW curves, with the latter being sampled just after rated speed, giving a sharp drop in thrust. The rated speed and maximum coefficient of power achieved by the models are in line with the assumptions made in the initial design, with the rotor designs reaching rated power at speeds within 0.2 m/s of 11.5 m/s, and within 0.05 of the design $C_P$ of 0.45.

The structural responses of the 100 kW and 1 MW blades are summarised in Figure 11. The figures are representative of the
structural responses of the blades under the same conditions for which the optimisation is carried out, namely DLCs 1.1 and 1.3. Turbulent wind files are generated using TurbSim for each tested windspeed — a coarse range between cut-in speed and cut-out speed — with the assumption that the inherent variation within the turbulent windfields results in adequate coverage of the operating range. These input files are provided in the associated repository for this paper. The piecewise linear aerodynamic model used for optimisation in ATOM is ill-equipped to handle the control-switching behaviour at rated wind (Macquart et al.,
2020). Hence, the performance of the rotors is analysed at windspeeds including 9 m/s and 13 m/s, but no wind field is generated



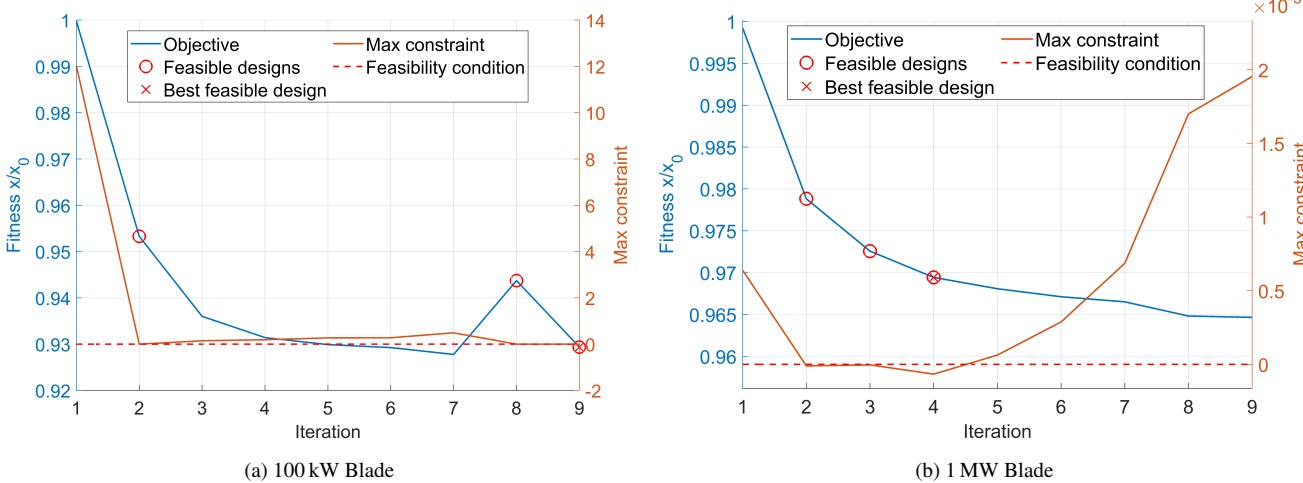

(a) 100 kW Blade

(b) 1 MW Blade

**Figure 8.** Optimisation histories for 100 kW and 1 MW blades. For a design to be feasible, the maximum constraint must be below zero — a threshold which is marked by a dotted red line. The feasible designs are marked on the blue objective curve with a red circle, and the feasible design with the lowest corresponding objective value is marked with a red cross.

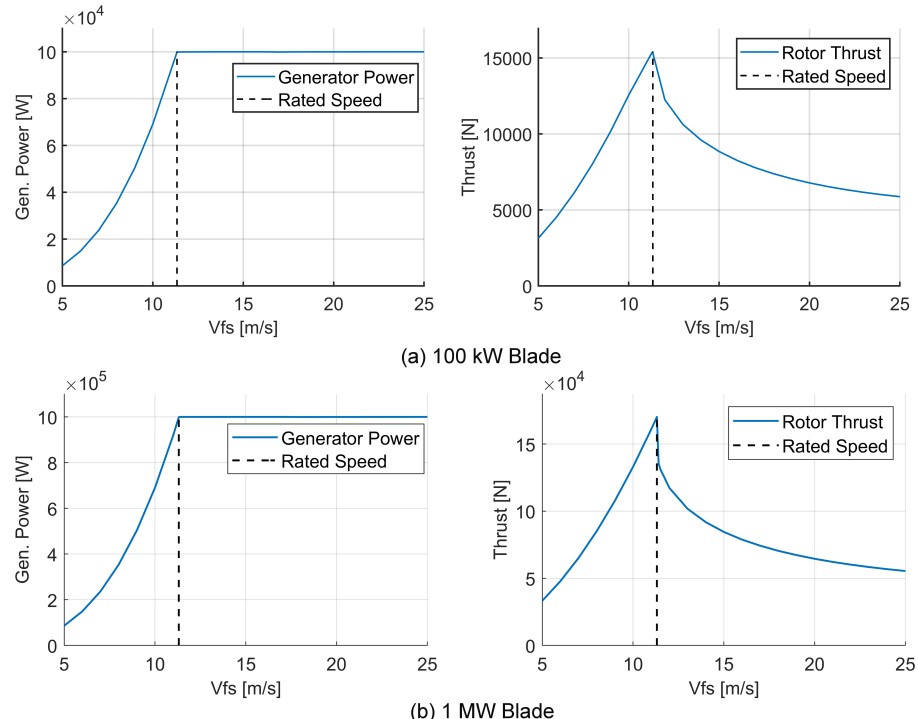

**Figure 9.** Power and thrust curves for (a) 100 kW and (b) 1 MW custom blade designs, as a function of freestream wind velocity (Vfs).





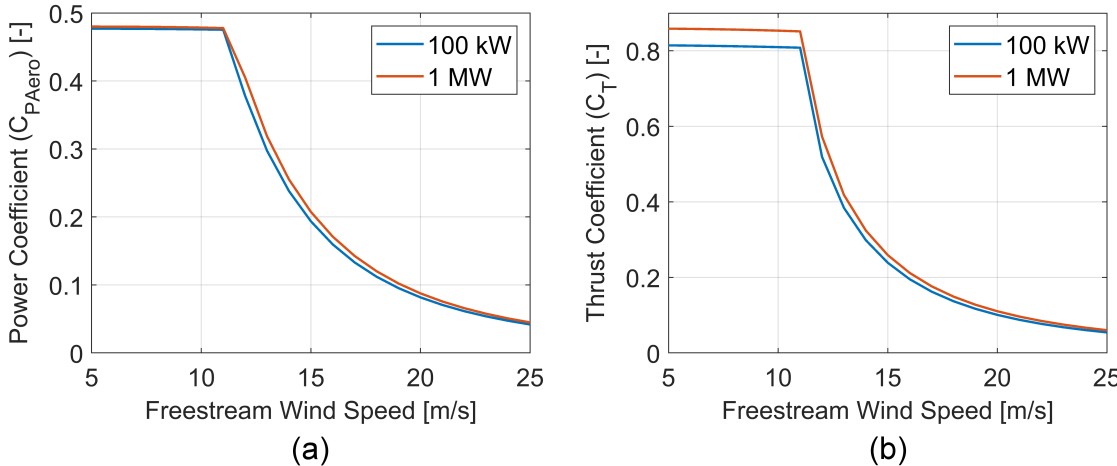

**Figure 10.** Dimensionless (a) power and (b) thrust coefficients of 100 kW and 1 MW custom blade designs, as a function of freestream wind velocity.

at an average hub speed equal to rated windspeed. This is assumed to sufficient in generating load envelopes for preliminary design, as adjustments can be made to the control scheme to limit loading at rated speed if needed (e.g., peak shaving). As expected, all windspeeds demonstrate a greater tip deflection and root bending moment for DLC 1.3, which simulates extreme turbulence. The resulting deflection and bending moment values are in line with published studies on 100 kW and 1 MW wind

turbine blades (Fuglsang and Madsen, 1995; Gözcü et al., 2022; K. Kim, H. G. Kim, and Paek, 2020). The ATOM controller, based on optimal power coefficient, is employed for the generation of these results. The structural behaviour is highly dependent on the controller settings, meaning that loading may be reduced through the testing of alternative controllers.

Figure 12 shows the material performance of the 100 kW and 1 MW blade designs using the sectional proximity to failure in buckling, strain and fatigue. The design appears quite conservative for the 100 kW blade, with failure indices remaining below

-0.6 for fatigue. In the ATOM optimiser, the values of constraints are scaled in order to give each constraint a similar order of magnitude for simpler comparison. The fatigue constraint value is quite strongly scaled, which is why the chosen design for the 100 kW blade appears very close to failure in Figure 8, but not in Figure 12. However, sections of the 1 MW blade experience loads that bring the design close to its material limit, especially in fatigue. This is consistent with the optimisation history of the 1 MW blade from Figure 8, which shows the difficulty faced by the optimiser in ensuring structural feasibility.

## 3.2  Aeroelastic Validation

In order to ensure the final open-source rotor models are representative of the rotors designed in ATOM, a campaign of code-to-code aeroelastic validation between ATOM and OpenFAST is carried out. As in previous sections, the 100 kW and 1 MW rotors are used as benchmarks for this validation, and the aeroelastic response of the blades in each program is compared under the same simulated conditions (dynamic aeroelastic simulations under uniform wind). The modal structural model in ATOM




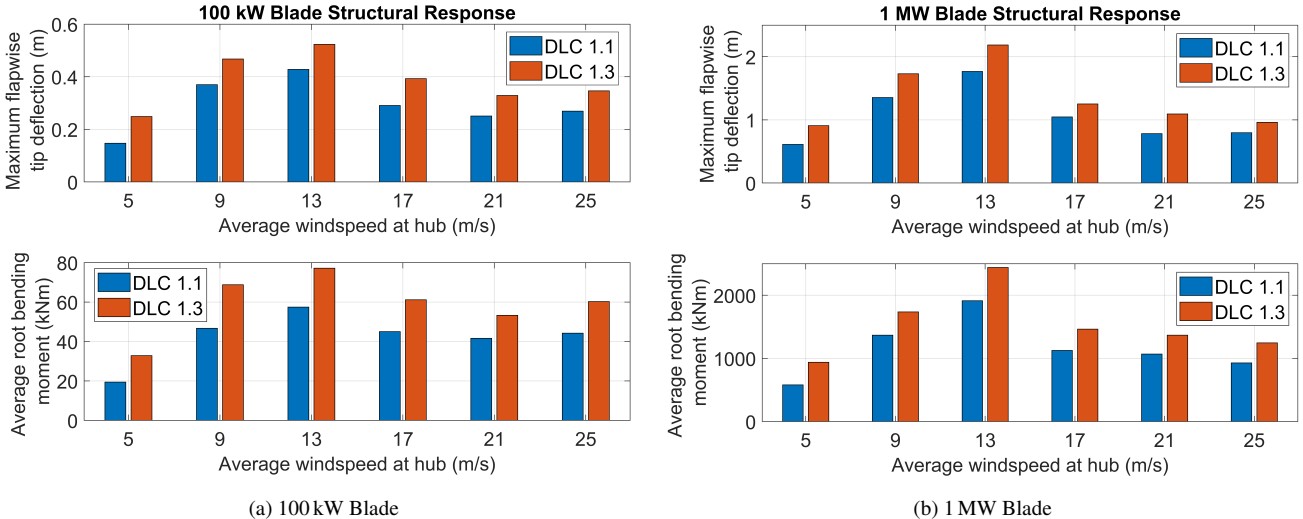

(a) 100 kW Blade   (b) 1 MW Blade

**Figure 11.** Maximum tip deflection and average root bending moment (both in the flapwise sense) of (a) 100 kW and (b) 1 MW custom blade designs, separated by average hub windspeed and DLC.

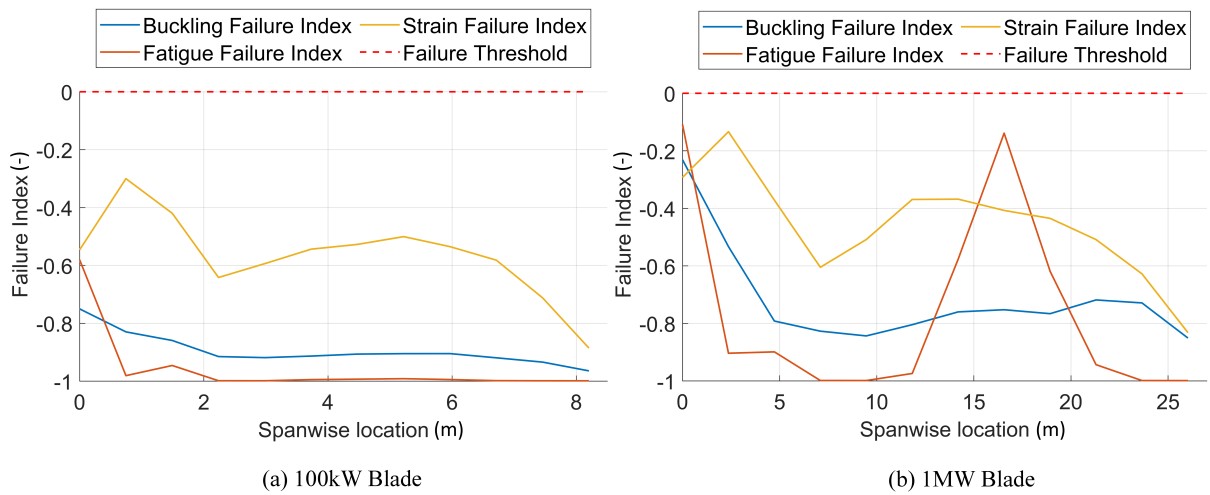

(a) 100kW Blade   (b) 1MW Blade

**Figure 12.** Maximum failure indices at each spanwise blade station for (a) 100 kW and (b) 1 MW rotor designs aggregated to show the maximum values from all simulations (windspeeds and design load cases).

differs in its implementation compared to the non-linear geometrically exact beam theory model used by the BeamDyn module of OpenFAST. These differences are likely the cause of the differing elastic response of the blades between the programs — a discrepancy that increases with greater windspeed. Figure 13 shows comparisons between ATOM and OpenFAST under uniform wind conditions for the operating range of windspeeds (cut-in to cut-out). The discrepancies in the structural models





used in ATOM and OpenFAST also appear larger for the smallest 100 kW rotor compared to the largest 1 MW rotor. A closer
look at the spanwise variation and time series behaviour for one windspeed can be found in Appendix A, Figure A1.

The differences in displacement propagate to the internal reaction loading envelopes, as ATOM uses the displacements to
calculate reaction loads. The root loading envelopes for each of the windspeeds shown in Figure 11 are included in Appendix
A, Figures A2 to A4. In each case, a 30-second quasi-static aeroelastic simulation is run in uniform wind conditions to assess
the results correlation between ATOM and OpenFAST in a simplified case.

**3.3   Rotor Interpolation**

The rotor library generated in this study is intended to enable greater flexibility in the modelling of multi-rotor wind turbines
for detailed design studies. Designing and comparing MRWT models under the constraint of solely using existing reference
designs is limiting with regards to the number and arrangement of rotors that can be explored for a given overall turbine rating.
For this reason, an interpolation procedure is implemented in ATOM to generate blade designs for intermediate ratings within
the 100 kW–1 MW reference rotor range. The function uses the piecewise cubic Hermite interpolation polynomial (pchip)
approach to interpolate a variety of blade parameters for the new design, including the blade length, twist, chord, and layup
thicknesses. Control parameters such as maximum rotor speed require additional checks post-interpolation to ensure tip speed
limits are not violated.

To validate the interpolated blade generation, a new blade design is generated at a rating of 375 kW using the interpolation
process, and its performance is compared against the equivalent optimised blade from the design library. In this case, the
optimised 375 kW design is removed from the library so the interpolation does not have access to it as a reference point.
Table 4 compares the mass, AEP and thrust at rated speed between the interpolated and optimised designs, showing adequate
similarity. These parameters are chosen as they are likely to be critical to the overall LCoE of the multi-rotor system through
their impact on the support structure sizing and the power generation. Figure 14 demonstrates the similarity between the
geometry of the interpolated blades, adding further confidence to the validity of the process in generating intermediate rotors
for multi-rotor design studies.

**Table 4.** Optimised 375 kW rotor parameters compared to the interpolated blade.

| Model | Blade Mass (kg) | AEP (GWh) | Thrust at rated speed (kN) |
|---|---|---|---|
| Optimised | 1046.7 | 1.3783 | 55.476 |
| Interpolated | 1153 | 1.3749 | 55.321 |

In order to better illustrate the use case of the rotor reference library, a selection of MRWT models is created. The overall
system rated power is set at 3 MW and the individual rotor capacities are determined based on the number of rotors. This
results in two designs that use the reference rotors described in this study directly — a 3-rotor model using three 1 MW rotors,
and a 15-rotor model using fifteen 200 kW rotors. These are the only structurally feasible configurations that can be generated
for a 3 MW multi-rotor system using rotors obtained solely from the reference library, and so this serves as an opportunity to





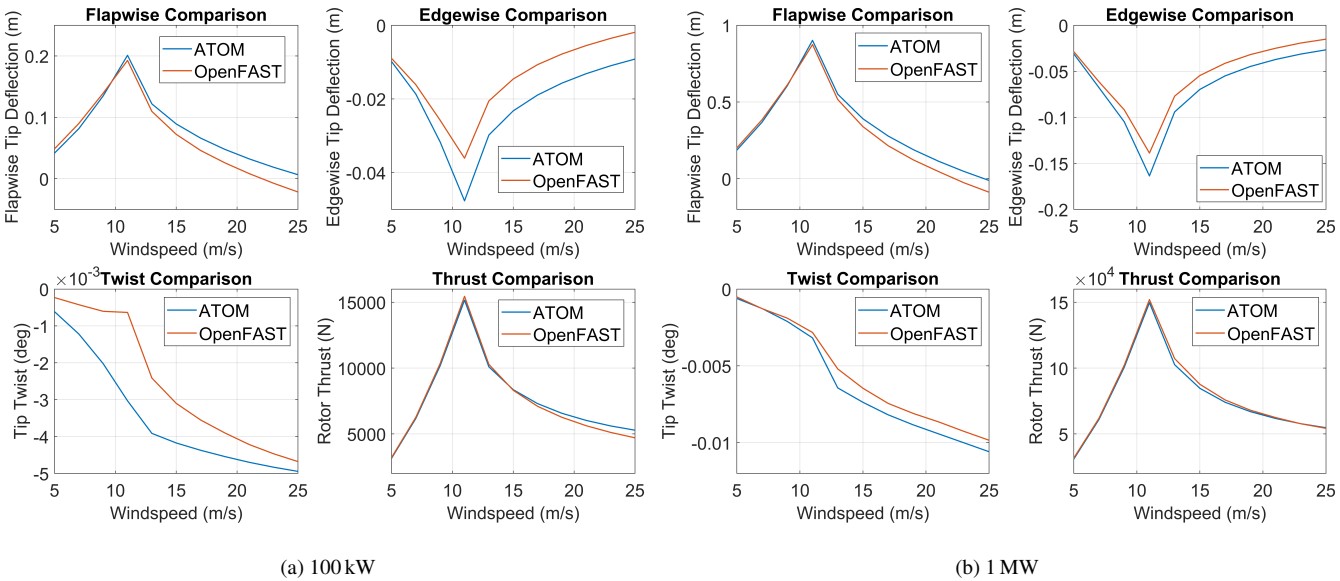

(a) 100 kW

(b) 1 MW

**Figure 13.** Structural deformations at the blade tip for (a) 100 kW and (b) 1 MW blade designs, compared between ATOM and OpenFAST. Rotor thrust is included to demonstrate the proximity in forces as opposed to the discrepancies in deflection, which suggests that the differing deflections are due to differences in the structural models.

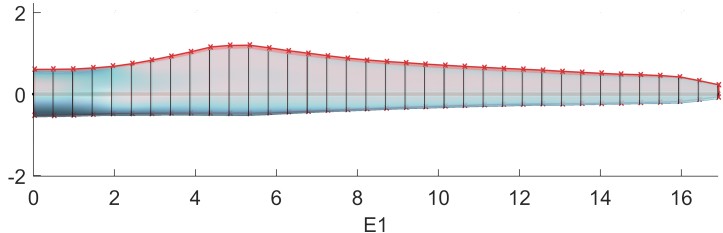

**Figure 14.** Outline of optimised 375 kW blade design (blue) overlaid with the interpolated design at the same power rating (red).

demonstrate the flexibility afforded by the interpolation process. Using the rotor library as a reference, an additional model is created using seven interpolated rotors at a rating of 428.6 kW each. These demonstrative models are displayed in Figure 15.

## 4 Discussion

A range of small reference rotors is devised in this study through aeroelastic optimisation using ATOM. These designs fill a gap in the wind turbine modelling field, which is the lack of available reference rotors below 750 kW and a valid means to interpolate between them. The mass trends for the rotors in the library are in keeping with the expected scaling of blade mass with length discussed in literature from historical precedent (Jamieson and Branney, 2012; Crawford, 2007). The comparisons

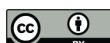

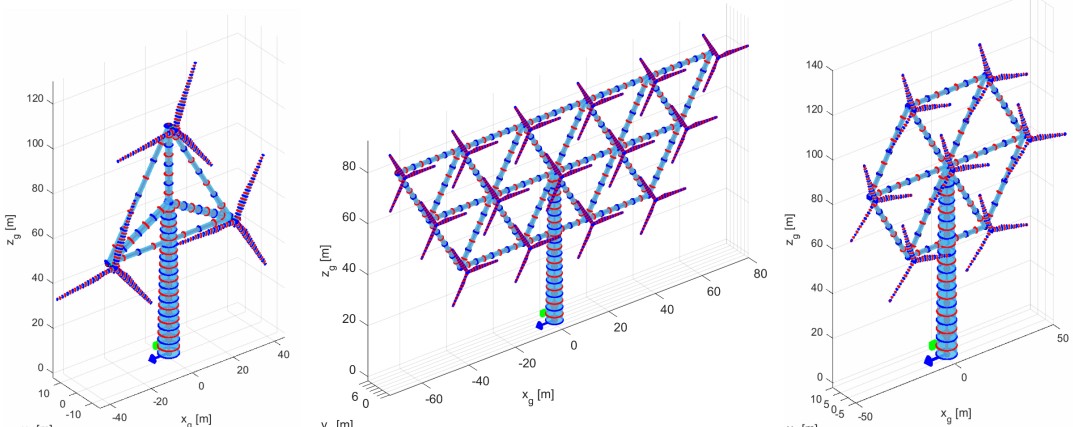

**Figure 15.** Multi-rotor models constructed using rotor designs from this study. These include three 1 MW rotors, fifteen 200 kW rotors and seven interpolated 428.6 kW rotors.

between the 750 kW blade design from this study and the WindPACT reference turbine demonstrate that the optimisation
process is able to converge to an optimised design within a relatively small number of iterations.

The fact that the WindPACT blade is still 24% lighter than the 750 kW blade may be of note. The WindPACT reference turbine report does not include detailed information on the design process of the blades, so the difference in methodology used for design is unclear. However, if the NREL aeroelastic tools were employed for its design, with similar load and deflection predictions to those generated by OpenFAST, it may be that the ATOM solver is conservative in comparison. This can be seen
from the consistent over-prediction of deflection in ATOM from Figure 13, which then translates to the internal loads. The ATOM-generated designs may also benefit from the optimiser being given greater design freedom. For example, the material layup was kept constant, with only the layer thicknesses as variables, and the chosen layup stacking sequence may not be optimal for the design scale being studied. A greater number of design variables, optimised for more iterations, with careful avoidance falling into local minima, may all contribute to a more optimal design for the blades. For this study, simplifications in
design variables and load cases are made to minimise the computational cost and timescale. The optimisation of the designs in the middle of the range (375 – 750 kW) faced difficulty in converging to their respective optimum designs, likely as a result of the optimiser converging to local minima instead of the global optimum. In each case, the resulting designs were significantly heavier than expected, not fitting the trend set by the other rotors in the library. This was addressed through the adjustment of the initial thickness distributions to values intentionally set below the structural performance limit in order to allow for the
optimiser to climb back up to a feasible point without over-designing.

The differences identified between ATOM and OpenFAST may also arise due to the programs applying blade pitching in different ways. In each simulation for the validation process (Figure 13), variable speed and pitch control are inactive, and control settings are fixed, being specified at the beginning of the simulation based on the mean uniform hub wind speed. Despite this, the discrepancies in the elastic response appear to diverge after rated speed, especially in the case of the flapwise

deflection. This implies the blade pitching may not be implemented in the same way in both programs. Nonetheless, external forces are predicted with good accuracy at all windspeeds except in the cases where the displacement difference is large, which suggests that the differences in structural model have a meaningful influence over the discrepancies in blade displacement.

## 5 Conclusions

The design processes and aeroelastic performance results for a library of reference wind turbine rotors are detailed in this study,
resulting in models of rated powers ranging from 100 kW to 1 MW. The aeroelastic responses of the blades are compared against those predicted by OpenFAST, with some discrepancies discussed in Section 4. In keeping with the stated purpose of the reference library, the rotor designs may be used to explore different rotor numbers and arrangements for multi-rotor wind turbine models. However, it may be of interest to pursue further refinement of the rotor designs in future work. Further optimisation using more advanced methods and a greater degree of design variable freedom, or the generation of high-fidelity
models based on those shared in this study are also potential routes of further research.

An area of ongoing research is the application of more environmentally sustainable materials, such as natural fibre or recycled composites, in wind turbine blades (Pender et al., 2024). The authors believe the small rotors detailed in this report may be suitable for further design studies on the implementation of sustainable materials in a wind energy context. MRWT research lacks detailed investigation of environmental impact, and research into the use of lower-impact materials in small rotor blades
may be an opportunity to deepen the understanding of the scientific community in this area.

*Code and data availability.* Reference rotor models are available in the GitHub repository at https://doi.org/10.5281/zenodo.17738642 (Sheik Hassan, 2025).

## Appendix A: Further Validation

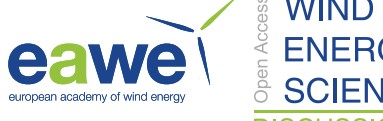

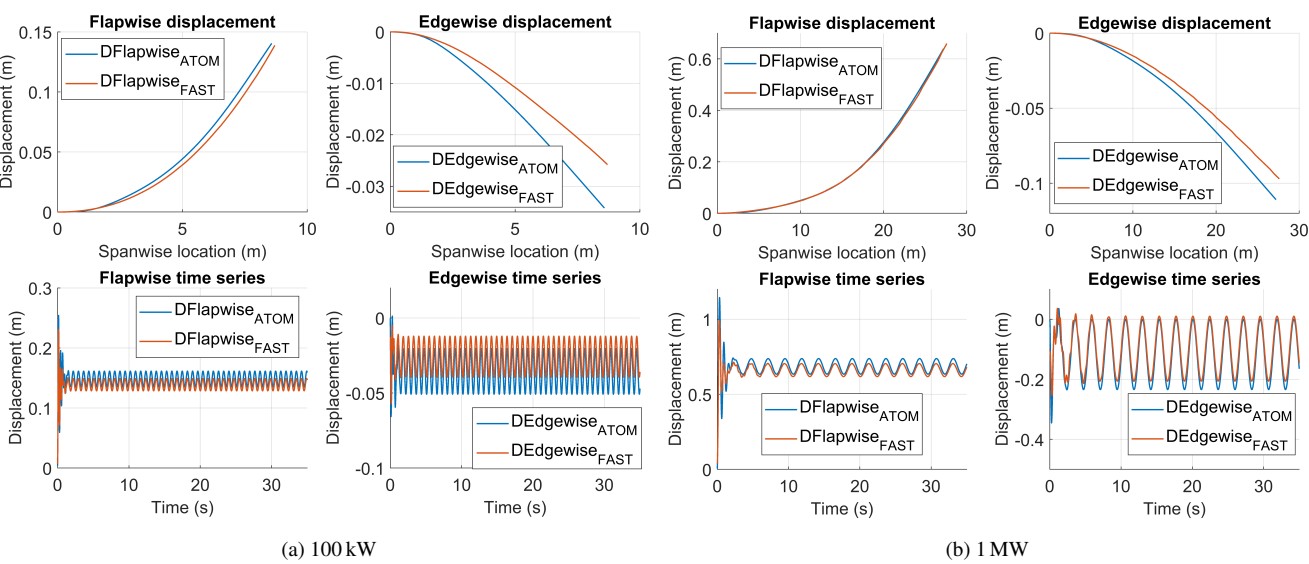

**Figure A1.** Elastic response of (a) 100 kW and (b) 1 MW blades in uniform 12 m/s wind, compared between ATOM and OpenFAST. In each case, the time-averaged displacements across the span as well as the time series for the tip displacements are compared.

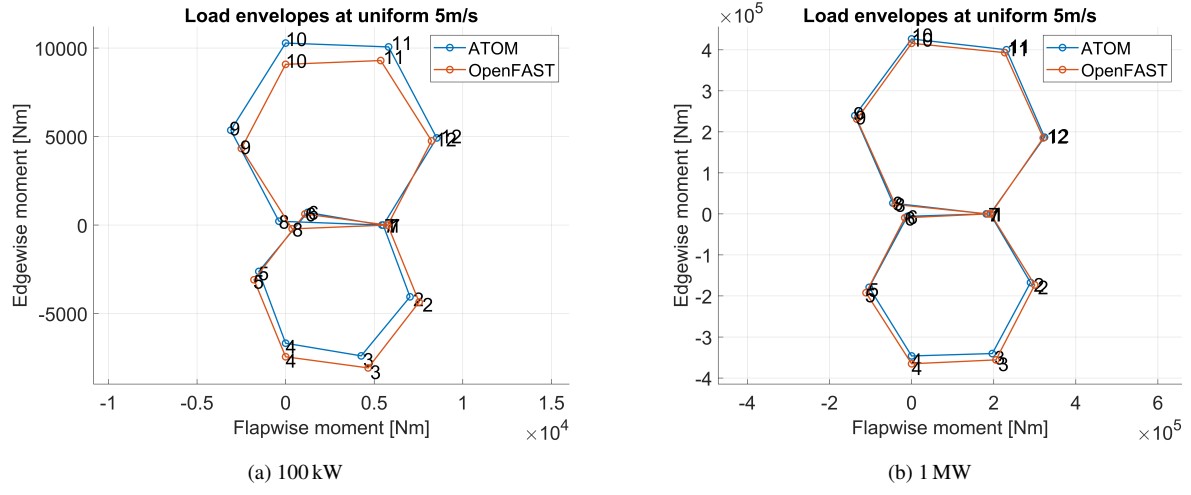

**Figure A2.** Load envelopes calculated at the blade root for the (a) 100 kW and (b) 1 MW reference blades in 5 m/s (cut-in) uniform wind conditions. To remove transient effects, the first 3 seconds of the 30 second simulation are omitted.





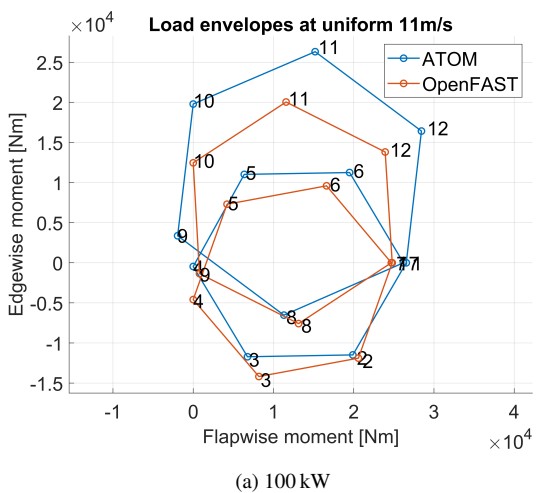

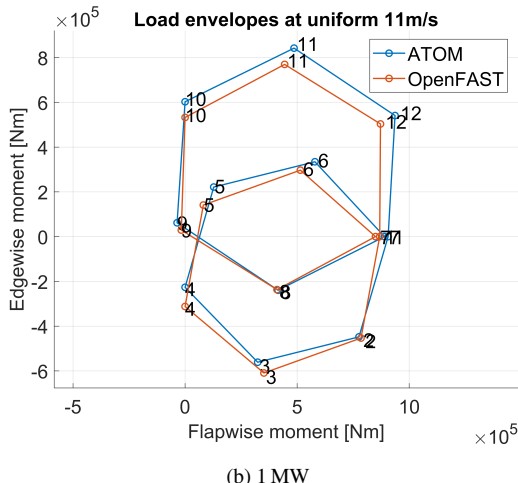

(a) 100 kW

(b) 1 MW

**Figure A3.** Load envelopes calculated at the blade root for the (a) 100 kW and (b) 1 MW reference blades in 11 m/s (near-rated) uniform wind conditions. To remove transient effects, the first 3 seconds of the 30 second simulation are omitted.

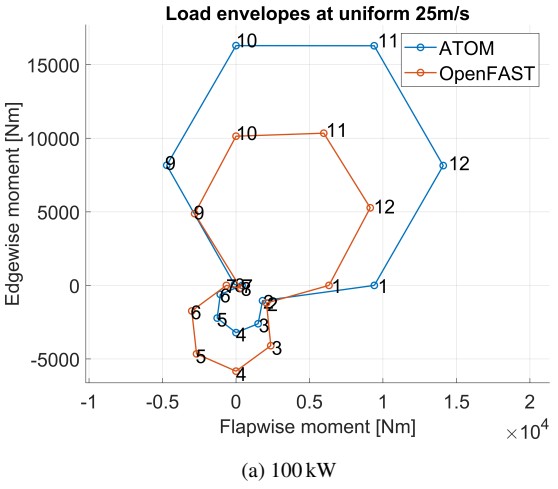

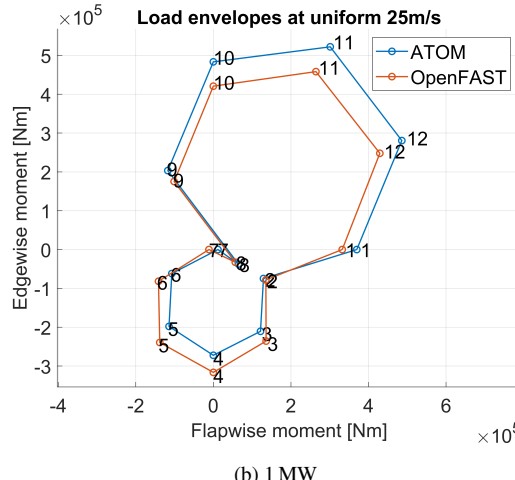

(a) 100 kW

(b) 1 MW

**Figure A4.** Load envelopes calculated at the blade root for the (a) 100 kW and (b) 1 MW reference blades in 25 m/s (cut-out) uniform wind conditions. To remove transient effects, the first 3 seconds of the 30 second simulation are omitted.



*Author contributions.* **AS**: Software, Methodology, Formal analysis, Writing - original draft, Visualization, Investigation, Funding acquisition. **NC**: Supervision, Writing - Review & editing, Funding acquisition. **RG**: Supervision, Writing - Review & editing, Funding acquisition. **TM**: Supervision, Conceptualization, Visualization, Methodology, Software, Writing - Review & editing, Funding acquisition.

*Competing interests.* The authors declare that they have no known competing financial interests or personal relationships that could have appeared to influence the work reported in this paper.

*Acknowledgements.* The authors would like to acknowledge the following organisations for their support throughout this project: (i) The funding of the Wind Blades Research Hub (WBRH), a joint collaboration between the University of Bristol and ORE Catapult, for contributing to the early development of ATOM (ii) EPSRC Doctoral Training Partnership (DTP) Studentship awarded to A. Sheik Hassan (iii) The Bristol Composites Institute at the University of Bristol.

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
