# Peer review of "Design optimisation of an open-source reference rotor library for multi-rotor development and innovation"

_Wind Energy Science, 2025_

## Referee Comment (RC1)

**Confidential Comments to the Editor:**

Thank you for the opportunity to review the Manuscript wes-2025-274 entitled " **Design optimisation of an open-source reference rotor library for multi-rotor development and innovation**" for the Journal of **Wind Energy Science**.

To the best of my knowledge, I have thoroughly reviewed the manuscript, and I would recommend in **revise the manuscript with minor comments** based on my review.

Thanks again for providing the opportunity to contribute the review support to maintain the quality standard of **Wind Energy Science** Journal.

I look forward to contributing more review support, so please share any additional manuscripts within my area of expertise. Thank you!

**Comments to the Author:**

As reviewer of the Manuscript wes-2025-274 entitled " **Design optimisation of an open-source reference rotor library for multi-rotor development and innovation**", I have thoroughly reviewed the manuscript, and I would recommend addressing the below comments to make the study more wholistic nature to appreciate the multi-rotor design optimization.

1. The global blade parameters play key role in designing the wind turbine rotor configuration. Hence the global blade parameters should be highlighted across both single-rotor vs multi-rotor configurations.
2. While optimizing the multi rotor system , the blade stiffnesses are plays critical role. So How this parameter is considered in the proposed optimization of multi-rotor concept models?
3. Blade prebend influences mainly in defining the tower clearance and accordingly designing the blade stiffness to avoid tower strike scenarios. Can you provide justification for not considering the blade prebend parameter?
4. How the optimization model behaves considering the blade prebend parameter?
5. It would be worth to add the material models considering in the multi rotor vs single rotor study configurations

I hope my critique helps the authors to improve their work and find useful in this review. Thank you!

---

## Referee Comment (RC2)

[referee-annotated manuscript omitted]